# Traditional versus Minimally Invasive Spinopelvic Fixation for Sacral Fracture Treatment in Vertically Unstable Pelvic Fractures

**DOI:** 10.3390/jpm12020262

**Published:** 2022-02-11

**Authors:** Yao-Tung Tsai, Yu-Ching Chou, Chia-Chun Wu, Tsu-Te Yeh

**Affiliations:** 1Department of Orthopaedic Surgery, Tri-Service General Hospital, National Defense Medical Center, No. 325, Sec. 2, Chenggong Rd. Neihu Dist., Taipei City 11490, Taiwan; andy27708092123@gmail.com (Y.-T.T.); doc20281@gmail.com (C.-C.W.); 2Division of Traumatology, Department of Surgery, Tri-Service General Hospital, National Defense Medical Center, No. 325, Sec. 2, Chenggong Rd. Neihu Dist., Taipei City 11490, Taiwan; 3School of Public Health, National Defense Medical Center, No. 161, Sec. 6, Minquan E. Rd., Neihu Dist., Taipei City 11490, Taiwan; trishow@mail.ndmctsgh.edu.tw; 4Medical 3D Printing Center, Tri-Service General Hospital and National Defense Medical Center, No. 325, Sec. 2, Chenggong Rd. Neihu Dist., Taipei City 11490, Taiwan

**Keywords:** minimally invasive technique, spinopelvic fixation, sacral fracture, pelvic ring fracture

## Abstract

Purpose: Numerous different fixation techniques are used to treat vertical shear sacral fractures. We report our experience with spinopelvic fixation using a minimally invasive technique. Methods: Thirty-eight patients with vertical pelvic and sacral fractures were treated with spinopelvic fixation (traditional open method, *n* = 21; minimally invasive technique, *n* = 17). Intergroup comparisons and statistical analysis were performed for intraoperative blood loss, operative time, post-operative radiographic grading, post-operative functional score, and complication rates. Results: Patients treated with the minimally invasive technique had a significantly shorter operative time (−52 min, *p* = 0.022), reduced blood loss volume (−287 mL, *p* < 0.001), and better cosmetic appearance (*p* < 0.05) than those in the traditional open group. There were no significant intergroup differences in post-operative radiographic grading (*p* = 0.489) or post-operative functional scores (*p* = 0.072). The complication rate was lower in the minimally invasive group (1/17 patients) than in the traditional open group (2/21 patients). Conclusions: Minimally invasive spinopelvic fixation is a viable treatment for sacral fractures and can reduce blood loss and operative time.

## 1. Introduction

The five sacral vertebrae, which have a triangular, concave shape, fuse to form the sacrum. The sacrum is a key stone of the axial skeleton and pelvic ring. The base of the sacrum articulates with the fifth lumbar spine and its intervertebral disc. The sacral ala articulates horizontally with the iliac bone and forms the sacroiliac joint, which is connected to strong ligaments. Biomechanically, the sacroiliac joint can afford axial loading from the trunk and transmit it to the lower limbs through the pelvic ring [1].

Sacral fractures show a bimodal distribution with high-energy trauma in young patients and low-energy trauma in elderly, metabolic, or neoplastic patients. The fracture pattern may be isolated or combined with pelvic ring fracture. Sacral fracture combined with anterior ring fracture is usually unstable and accounts for approximately 30–40% of all pelvic ring fractures [2]. Sacral fracture caused by high-energy trauma has a high incidence of associated injury and often a complex fracture pattern, which is difficult to interpret. According to the Denis classification [3], sacral fractures can be classified into three zones: zone I, the fracture line is lateral to the foramen and traverses the sacral ala; zone II, the fracture is transforaminal; and zone III, the fracture is medial to the foramen and traverses the central spinal canal.

Treatment of sacral fractures is debated among traumatic surgeons, with no clear consensus. The most common treatment methods for sacral fractures are posterior pelvic fixation (sacroiliac or transiliac-transsacral screws) and spinopelvic or triangular fixation [4]. Several reports have revealed that traditional posterior pelvic fixation methods such as the use of sacroiliac screws or transiliac plates are biomechanically less stable than spinopelvic fixation, which consists of pedicle screws from L4 or L5 connected by rods to pelvic fixation [5,6]. However, the traditional open spinopelvic fixation constructs also have disadvantages and complications such as a relatively high infection rate (10–15%), wound dehiscence, and instrument problems [4,7,8]. Two reports have demonstrated advantages of the minimally invasive technique of spinopelvic fixation compared to the traditional open method [9,10]. This study aimed to analyse and compare the clinical results of sacral fracture treatment with the traditional open method of spinopelvic fixation and the minimally invasive technique, and we hypothesized that the minimally invasive technique would demonstrate a better clinical outcome than the traditional open method.

## 2. Materials and Methods

We performed a retrospective analysis of patients with sacral fractures combined with vertically unstable pelvic fractures who underwent traditional or minimally invasive spinopelvic fixation at the Tri-Service General Hospital.

A total of 38 patients who had sustained fractures of the sacrum and underwent surgical treatment from June 2012 to May 2019 were included. Their medical records were collected for statistical analysis and their radiographic images were assessed. The inclusion criterion was a diagnosis of sacral fracture combined with a vertically unstable pelvic fracture. The exclusion criteria were age < 20 years, tumour in the pelvic or sacral region, associated lower limb neurovascular injury, and use of a transiliac plate. The patients were divided into two groups: the traditional open spinopelvic fixation group (group 1) and the minimally invasive spinopelvic fixation group (group 2). In group 1, 4 patients had isolated pelvic injury and 17 patients had accompanying limb fracture or other associated injury such as urologic injury, pneumothorax, rectal injury, initial hypovolemic shock, abdominal injury, intracranial haemorrhage, or chest injury. In group 2, 3 patients had isolated pelvic injury and 14 patients had accompanying limb fracture or other associated injury such as urologic injury, rectal injury, Morel-Lavallée lesion (internal degloving injury), chest injury, initial hypovolemic shock, or head injury. All included patients had undergone a pelvic anteroposterior view radiography and computed tomography (3 mm axial slices) for detailed preoperative examination. The Denis classification system was used to classify the sacral fractures.

Approval was obtained from the Institutional Review Board of the Tri-Service General Hospital (Approval no.: A202005132). Informed consent was obtained from all the patients included in the study.

### 2.1. Surgical Technique

All surgeries in both groups were performed by the same orthopaedic pelvic trauma surgeon of our department with 20 years of experience. All patients underwent surgery in the prone position on a radiolucent table. In group 1, the traditional midline open approach was used. Pedicle screws (SmartLoc Spinal Fixation System, A-SPINE ASIA CO., LTD., Taipei, Taiwan) were inserted into the L5 pedicles and the iliac bone under fluoroscopy. The connecting rod was bent to fit the screws’ location and was fixed with nuts. In group 2, a Jamshidi needle (Becton, Dickinson and Company, Franklin Lakes, NJ, USA) was inserted through a small paramedian stab incision. The Jamshidi needle was docked on the lateral pedicle margin, midway between the superior and inferior aspects of the pedicle, under fluoroscopic guidance. Fluoroscopic images were obtained each time the needle was advanced. After the position of the pedicle screw was confirmed with fluoroscopy, the Jamshidi needle was removed, and a guidewire was inserted into the pedicles to guide the pedicle screws (6.5 mm in diameter, 45–50 mm in length) (CD HORIZON^®^ SEXTANT^®^ II Spinal System, Medtronic, MN, USA). The same technique was used with the iliac screw (6.5 mm in diameter and 55 mm in length), wherein the entry point was located at the recess below the posterior iliac spine and the Jamshidi needle was used to penetrate the cortical bone at 45° in both the ventral and caudal directions toward the anterior inferior iliac spine for purchasing the thickest bone above the greater sciatic notch. The connecting rod was percutaneously applied using the Sextant rod insertion system (Figure 1). The final fixation construct was illustrated in Figure 2.

### 2.2. Outcome Measurement

Post-operative image assessment was performed using the standard pelvic radiographic series (anteroposterior, lateral, inlet, and outlet views). The Iowa pelvic score was used to evaluate postoperative functional scores 6 months after surgery [11]. Follow-up radiographs showed fracture reduction quality and screw position. After discharge, all patients were regularly followed up at the outpatient department at 2 weeks, 4 weeks, 8 weeks, 3 months, 6 months, and 1 year. All postoperative image studies were assessed by three different orthopaedic trauma surgeons, the fracture reduction quality was assessed, and a consensus was achieved. The results of the sacral fracture reduction quality were classified into three different grades: excellent (0–5 mm displacement), good (5–10 mm displacement), and fair (10–20 mm displacement). Total operative time, blood loss volume, and fracture reduction quality were assessed and statistically analysed. Total operative time was recorded from skin incision to the end of complete skin closure. Intraoperative blood loss volume was recorded in the medical chart. Complications included wound infections, soft tissue impingement by implants, implant failure, and implant loosening.

A normality check was conducted using Shapiro–Wilk’s test. Data are presented as the mean ± standard deviation for continuous variables, and we compared the differences between groups using independent *t*-tests. Categorical variables are reported as numbers and percentages (%), and we compared the differences between groups using chi-square tests, or Fisher’s exact tests if the expected numbers in any cell were less than five. We ordered α = 0.05, *n* = 21 in group 1 and n = 17 in group 2, and then used two-tailed independent *t*-tests to calculate the power (1 − β) = 61.9%, 100%, and 44.9% for operation time, blood loss, and Iowa score, respectively. All statistical tests were two-tailed, and *p* < 0.05 was defined as statistically significant. The Statistical Package for the Social Sciences software (version 22.0, IBM Corporation, Somers, NY, USA) was used for all statistical analyses.

## 3. Results

Data including age, sex, Denis classification, days from injury to surgery, unilateral or bilateral sacral involvement, use of sacroiliac screws, associated injury, and fixation of the anterior pelvic ring according to group are shown in Table 1. Overall, no significant difference was noted in the clinical demographic data between the two groups. The average follow-up duration was 30.6 months (range, 10–75 months) (Table 1). No patient was lost to follow-up.

### 3.1. Perioperative Clinical Parameters

The total operative time and blood loss volume were significantly different between groups 1 and 2 (193.19 ± 89.88 vs. 141.47 ± 34.88 min, *p* = 0.022; 330.48 ± 137.20 vs. 42.65 ± 25.50 mL, *p* < 0.001; Table 2).

### 3.2. Postoperative Radiographic Results and Functional Outcome

The pre- and postoperative radiographs in the traditional and minimally invasive spinopelvic fixation are shown in Figure 3 and Figure 4, respectively.

The post-operative radiographic grading in groups 1 and 2 was excellent, good, and fair in 15, 5, and 1 cases and 9, 7, and 1 cases, respectively. The IOWA pelvic score was 83.3 ± 8.1 and 87.6 ± 5.7 in groups 1 and 2, respectively. Group 2 had better functional scores than group 1 but without statistical significance (*p* = 0.072) (Table 3). However, associated injury may affect the performance of the IOWA score. We analysed the IOWA scores in each group with or without an associated injury. Patients with and without associated injury showed no significant difference between the groups (Table 3). As for cosmetic appearance, group 2 patients had significantly better cosmetic effects after surgery (*p* = 0.007) (Table 3).

### 3.3. Postoperative Complications

Two group 1 patients had complications: one had a broken pedicle screw when the implant was removed 9 months postoperatively, and the other had pedicle screw loosening and was treated with revision surgery and prolonged protected weight-bearing with crutches for 6 months. The fracture site healed without any discomfort. In group 2, one patient had a surgical wound infection over the iliac crest region after anterior pelvic ring fixation. However, the surgical wound over the back was clean. After surgical debridement and antibiotic treatment, the patient recovered, and the fracture site healed 6 months later. No implant irritation or nonunion was observed in either group.

## 4. Discussion

There is no clear consensus regarding surgical techniques for sacral fracture treatment, and the successful treatment for these sacral fractures remains a challenge to orthopaedic trauma surgeons because of its complex anatomy and fracture pattern. Sacral fractures are often accompanied with other associated injuries such as pelvic ring fractures, lower limb fractures, other axial bone fractures, or visceral injuries [1,9]. Shorter operative time and less intraoperative blood loss volume may prevent haemodynamic instability and reduce subsequent systemic complications. Extensive surgical dissection to expose the posterior structures of the lower lumbar spine and posterior iliac crest is necessary in the traditional open method. This may cause muscle stripping, which may denervate the paraspinal musculature and increase intraoperative blood loss [9]. In addition, the traditional midline open approach requires considerable time. In the minimally invasive technique, a small incision of 2.5–3 cm is made lateral to the midline. A Jamshidi needle is inserted into the lumbar pedicle and iliac bone under fluoroscopic guidance without stripping or detaching the muscles. Consistent with our results, many reports have revealed that the minimally invasive technique can save considerable time and reduce blood loss, with a significant difference compared to the traditional open method [4,10,12]. According to König et al. [13] and Bellabarba et al. [8], higher infection and wound dehiscence rates were noted with traditional open spinopelvic fixation than with minimally invasive spinopelvic fixation. Koshimune et al. [9] reported decreased operative time and intraoperative blood loss volume, low postoperative infection rates, and low amounts of rigid fixation needed for a high rate of bony union in the minimally invasive group. Prolonged operative time, large surgical wound, and muscle stripping may have led to this result.

The minimally invasive method has some limitations in fracture reduction compared to the open method. The traditional open method allows variable open reduction and direct reduction with clamps or Shanz screws as levers [12,14]. In our study, the minimally invasive method entailed indirect reduction using the rod-distraction technique [15], and the manual method entailed leg traction with counter force over the patient’s shoulder. Preoperative distal femur skeletal traction was also performed to prevent further displacement. The postoperative radiographic grading and union rate showed no significant difference, and it revealed that the minimally invasive technique provided similar reduction quality and better surrounding tissue and vascular restoration, which promoted better bony healing.

Bellabarba et al. [8]. reported that approximately 11% of patients in the traditional open group underwent reoperation because of implant irritation. In our study, neither group reported implant irritation. We used the modified iliac screw fixation technique with the entry point located at the recess below the posterior superior iliac spine with the direction toward the anterior inferior iliac spine.

This technique has two important advantages: (1) it significantly decreases screw head irritation and (2) a connector rod is easy to apply without the necessity of offsetting the connecting rod, which can save operative time [16]. However, we removed implants in both of the groups 6 months after surgery to prevent pedicle screw damage. One patient in group 1 had broken pedicle screws at L5. The spinopelvic fixation of the L5 to the iliac bone had only pedicle screws and connecting rods without any fusion procedure. The movement of L5 and the sacrum may lead to broken pedicle screws or rods [17].

Sagi [18] and Kaye [19] reported that spinopelvic fixation combined with L4, L5, and iliac bone provides rigid stability for vertical shear sacral fractures. In our study, all patients in both groups underwent L5 to iliac spinopelvic fixation, except for one patient who underwent L4, L5, and iliac bone construct because of osteoporotic bone quality. In the study by Sagi [18], the iliac screw was at least 100 mm long for solid bony purchase. In our study, the iliac screw used in group 2 was 6.5 mm in diameter and 55 mm in length. In our trajectory of the iliac screw, the iliac screws were inserted into the sciatic buttress to purchase the thickest bone and provide solid stability. All patients achieved bony union without any loss of reduction. The L5 to ilium spinopelvic fixation may provide efficient stability for vertical shear sacral fractures; however, a biomechanical study is needed to obtain further evidence.

The postoperative mid-term functional outcomes were measured using the Iowa pelvic score in this study. However, there was no significant difference in the total Iowa pelvic scores. This means that this minimally invasive spinopelvic fixation technique and the traditional open technique had similar results. However, the cosmetic appearance in the Iowa pelvic score was significantly different between the two groups. The minimally invasive group showed better results, indicating its value.

In our county, due to the different cost between theses implants, the pedicle screw and rod implantation by the minimally invasive method is more expensive and is paid by the patient. That said, the pedicle screw and rod implantation by the open method could be supported by the National Health Insurance Administration after written application, which should be reviewed and qualified by the expert. In this study, the implants and the surgical method were decided by the patients’ economic conditions. This was the reason why these patients could not be randomly selected. Limitations of our study included having a relatively small group of patients and that this study was a retrospective study. In the future, a similar study can include larger case numbers combined with the randomized control trial model; this would be much more convincing and provide a higher evidence level.

In conclusion, the minimally invasive spinopelvic fixation technique is a safe and effective treatment for sacral fractures. Most complications noted with the traditional open spinopelvic fixation method, as mentioned in previous reports, could have been prevented by using the minimally invasive technique. This new technique showed shorter operative time and reduced intraoperative blood loss and had a better cosmetic effect than the traditional method.

## Figures and Tables

**Figure 1 jpm-12-00262-f001:**
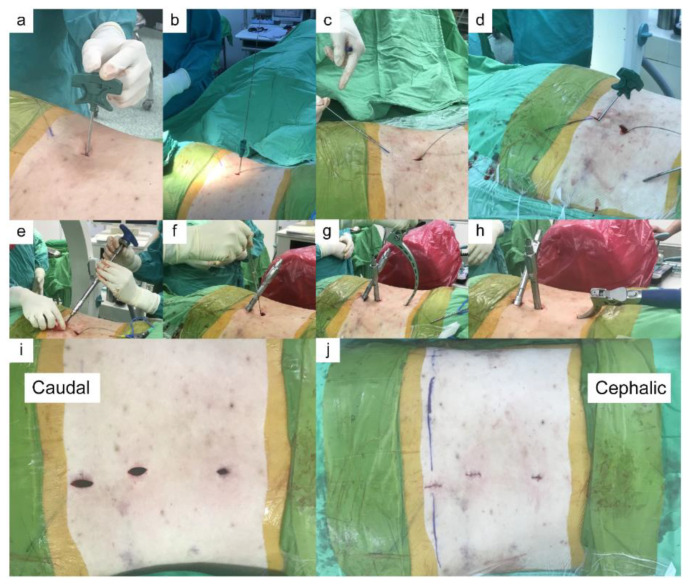
(**a**) The entry point of the pedicle screw is determined using a Jamshidi needle under fluoroscopic guidance. (**b**) The Jamshidi needle is inserted into the pedicle, and a blunt guide wire is inserted through the Jamshidi needle. (**c**) The direction of iliac screw is determined with a Steimann pin under fluoroscopic guidance and a guiding line is drawn on the skin. (**d**) The Jamshidi needle is docked into the recess below the posterior iliac spine following the guiding line at 45° in the ventral and caudal directions toward the anterior inferior iliac spine. The blunt guiding pin is placed through the Jamshidi needle. (**e**) The cannulated iliac screw is inserted. (**f**) The L5 pedicle screw is inserted. (**g**,**h**) The Sextant II percutaneous rod system is used to deliver the rod percutaneously. (**i**,**j**) The surgical wound of the minimally invasive technique.

**Figure 2 jpm-12-00262-f002:**
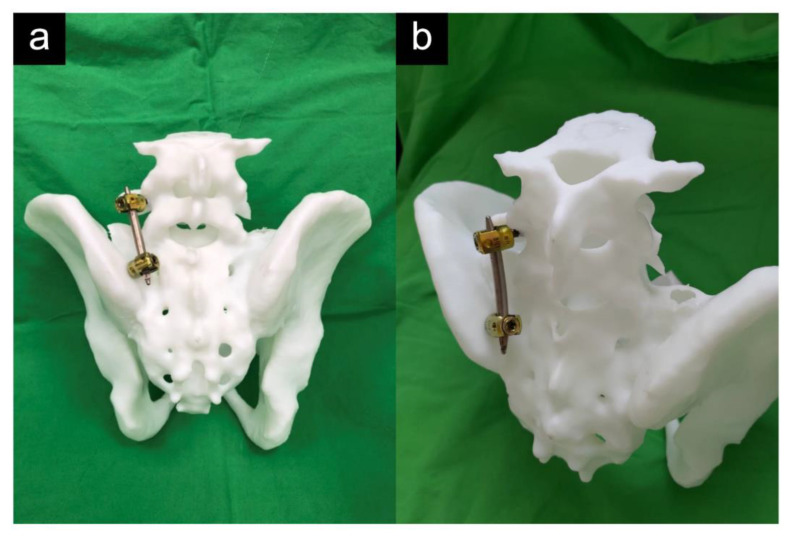
Minimally invasive spinopelvic fixator in three-dimensional printing model. (**a**) Posterior view of the spinopelvic fixator. (**b**) Oblique view of the spinopelvic fixator and the entry point of iliac screw located at the recess below the posterior iliac spine. The screw head can hide under the bony prominence. This 1:1 3D printing bone model is supported by the Medial 3D Printing Center, Tri-Service General Hospital, National Defense Medical Center, Taipei, Taiwan.

**Figure 3 jpm-12-00262-f003:**
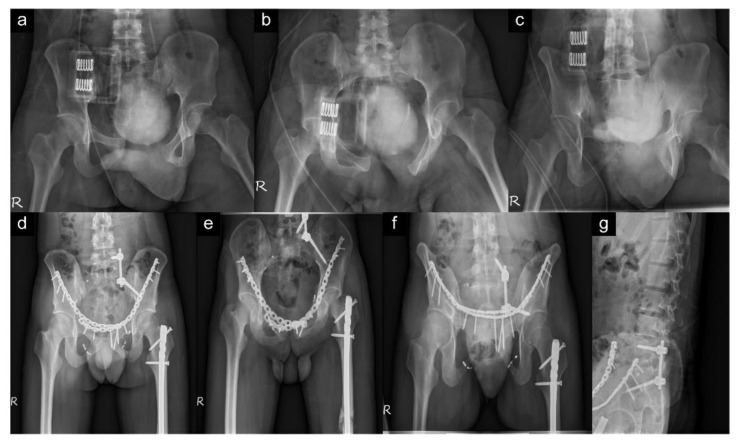
A 21-year-old man with anterior and posterior pelvic ring injury. The “R” on the plane radiography represents the right side of the patient. (**a**–**c**) Preoperative pelvic radiology series (anteroposterior, inlet, and outlet views) showing sacral fracture, Denis zone II, left, and diastasis of pubic symphysis with right superior and inferior pubic rami and left inferior pubic ramus. (**d**–**f**) Postoperative pelvic radiology series (anteroposterior, inlet, and outlet views) demonstrating traditional spinopelvic fixator and dual plating for anterior pelvic ring fixation. Antegrade intramedullary nailing for femoral shaft fracture (left) is also noted. (**g**) Postoperative sacral lateral view showing good position of spinopelvic fixator of the L5 and ilium.

**Figure 4 jpm-12-00262-f004:**
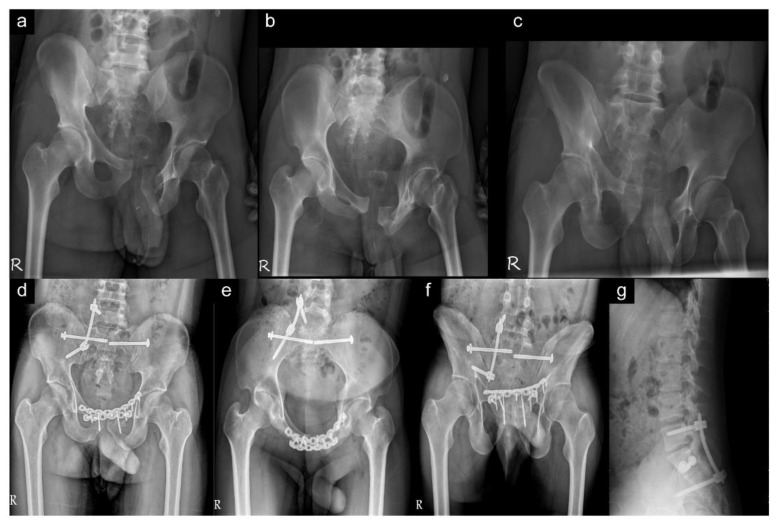
A 27-year-old man with anterior and posterior pelvic ring injury. The “R” on the plane radiography represents the right side of the patient. (**a**–**c**) Preoperative pelvic radiology series (anteroposterior, inlet, and outlet views) showing right sacral fracture (Denis zone II) with diastasis of the bilateral sacroiliac joint and of pubic symphysis with fracture of the left anterior pubic ramus. (**d**–**f**) Postoperative pelvic radiology series (anteroposterior, inlet, and outlet views) demonstrating bilateral percutaneous sacroiliac screws, minimally invasive spinopelvic fixator, and dual plate fixation for anterior pelvic ring. (**g**) Postoperative lumbar and sacral lateral view showing good position of sacroiliac screw and spinopelvic fixator of the L5 and ilium.

**Table 1 jpm-12-00262-t001:** Patients’ demographic data.

Variables	Group 1 (*n* = 21)	Group 2 (*n* = 17)	*p* Value
Age, M ± SD	38.24 ± 12.91	44.71 ± 19.07	0.222 ^a^
Day to surgery, M ± SD	9.38 ± 7.36	7.71 ± 5.99	0.454 ^a^
Sex			1.000 ^a^
Female	9 (42.9%)	7 (41.2%)	
Male	12 (57.1%)	10 (58.8%)	
Denis classification			0.101 ^b^
Zone I	8 (38.1%)	12 (70.6%)	
Zone II	12 (57.1%)	5 (29.4%)	
Zone III	1 (4.8%)	0 (0%)	
Implant			0.638 ^a^
Unilateral	15 (71.4%)	10 (58.8%)	
Bilateral	6 (28.6%)	7 (41.2%)	
Sacroiliac screw			1.000 ^b^
No	5 (23.8%)	4 (23.5%)	
Yes	16 (76.2%)	13 (76.5%)	

M ± SD: Mean ± standard deviation, ^a^
*t*-test or chi-square test, ^b^ Fisher’s exact test.

**Table 2 jpm-12-00262-t002:** Surgical outcome analysis.

	Group 1 (*n* = 21)	Group 2 (*n* = 17)	*p* Value
Operation time	193.19 ± 89.88	141.47 ± 34.88	0.022 ^a^
Blood loss	330.48 ± 137.20	42.65 ± 25.50	<0.001 ^a^
Radiographic grading			0.489 ^b^
Excellent	15 (71.4%)	9 (52.9%)	
Good	5 (23.8%)	7 (41.2%)	
Fair	1 (4.8%)	1 (5.9%)	
Complications			1.000 ^b^
No	19 (90.5%)	16 (94.1%)	
Yes	2 (9.5%)	1 (5.9%)	

M ± SD: Mean ± standard deviation, ^a^
*t*-test or chi-square, ^b^ Fisher’s exact test.

**Table 3 jpm-12-00262-t003:** Iowa score and cosmetic appearance.

	Group 1 (*n* = 21)	Group 2 (*n* = 17)	
	No. (%)	No. (%)	^a^*p*-Value
Iowa score, M ± SD	83.3 ± 8.1	87.6 ± 5.7	0.072
With associated injury	(*n* = 17)	(*n* = 14)	
Iowa score, M ± SD	82.8 ± 7.8	87.6 ± 5.9	0.067
Without associated injury	(*n* = 4)	(*n* = 3)	
Iowa score, M ± SD	85.3 ± 10.2	87.3 ± 5.1	0.763
Associated injury			1.000
Yes	17 (81.0)	14 (82.4)	
No	4 (19.0)	3 (17.6)	
Cosmetic appearance			0.007
Significant	14 (66.7)	3 (17.6)	
Not significant	7 (33.3)	14 (82.4)	

M ± SD: Mean ± standard deviation, ^a^
*t*-test or chi-square test.

## Data Availability

All relevant data are within the manuscript files.

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
