# Peer review of "Traditional versus Minimally Invasive Spinopelvic Fixation for Sacral Fracture Treatment in Vertically Unstable Pelvic Fractures"

_jpm, 2022, doi:10.3390/jpm12020262_

Round 1

Reviewer 1 Report

Tsai et al presents an interesting paper with a very important topic. The paper is decently written and the presentation of the data is adequate.

However, few points remain unclear.

1- the study is descriptive of patients admitted to the institute of the authors at best, the description as a retrospective study is very ambitious. This is based on the low sample number which should also be discussed as a limitation of the study.

2- The statistical analysis is not justified, why use Fisher's exact test? as this test is used to determine whether the association between to variables is random or not. this is not reflected in the results as the authors report significant differences and not associations. 2- Power analysis for the response variables in table 2 alone with a medium effect size to use Fishers suggest the need of 151 patients per group at least.

Please provide your power analysis that justified the number of patients or describe as a descriptive study.

3- The unclear use (when or why) of Chi square or T-test s very confusing.  please explain.

4- please add what test was used to determine normality of data.

5- The authors are encouraged to discuss their results in light of the short and long term follow-up prediction based on other reports example PMID: 31380389, PMID: 27847854. The paper addresses cases from 2012 to 2019 however the follow-up period was not clear.

6- is there any overlap in these patients and those of Tsai et al, 2019 (PMID: 30958869). If so authors shall indicate this clearly in this manuscript.

Author Response

Tsai et al presents an interesting paper with a very important topic. The paper is decently written and the presentation of the data is adequate.

However, few points remain unclear.

1- the study is descriptive of patients admitted to the institute of the authors at best, the description as a retrospective study is very ambitious. This is based on the low sample number which should also be discussed as a limitation of the study.

Response: Thank you for your comment. The limitation of this study is not a randomized control trial study and limited case number. Please see the revised manuscript on line 283-286.

2- The statistical analysis is not justified, why use Fisher's exact test? as this test is used to determine whether the association between to variables is random or not. this is not reflected in the results as the authors report significant differences and not associations. 2- Power analysis for the response variables in table 2 alone with a medium effect size to use Fishers suggest the need of 151 patients per group at least.Please provide your power analysis that justified the number of patients or describe as a descriptive study.

Response: Thank you for your comment. We used Fisher's exact tests to compare the differences of categorical variables (ex: radiographic grading and complications) between groups 1 and 2 if the expected numbers in any cell were less than five. In addition, we used independent t tests to compare the differences of continuous variables (ex: operation time and blood loss) between groups 1 and 2. Moreover, we ordered α=0.05, n=21 in group 1 and n=17 in group 2, then we used independent t tests with two tailed to calculate the power(1-β)= 61.9%, 100% and 44.9% for operation time, blood loss and Iowa score. Please see the revised manuscript on line 139-147.

3- The unclear use (when or why) of Chi square or T-test s very confusing.  please explain.

Response: Thank you for your comment. Please see the revised manuscript on line 139-143.

4- please add what test was used to determine normality of data.

Response: Thank you for your comment. We have added “A normality check was conducted using the Shapiro–Wilk’s test”. Please see the revised manuscript on line 139.

5- The authors are encouraged to discuss their results in light of the short and long term follow-up prediction based on other reports example PMID: 31380389, PMID: 27847854. The paper addresses cases from 2012 to 2019 however the follow-up period was not clear.

Response: Thanks for your comment. In our study, all the patient was regularly followed at OPD until post-operative 6 to 9 months when we could see bony union on the X ray film and the second operation of implant removal was arranged. After the implant was removed, we did not keep following these patients. Thanks for your precious opinion and suggestion, in the future study, we could follow up these patients for a longer period to discuss the short-term and long-term follow-up result.

6- is there any overlap in these patients and those of Tsai et al, 2019 (PMID: 30958869). If so authors shall indicate this clearly in this manuscript.

Response: Thanks for your reminder, there is 4 patients in the traditional open method group were overlapped with those patients of Tsai et al, 2019 (PMID: 30958869). All these 4 cases had vertical shear sacral fracture with anterior pelvic ring fracture. In this study, we focused on the posterior fixation and the previous study we discussed about anterior pelvic ring fixation. These two studied had no obvious relevance.

Reviewer 2 Report

An interesting article, a topic that has not been fully understood so far.

However, I have a few comments:

INTRODUCTION: Please add a research hypothesis. MATERIALS AND METHODS: Please state how many patients in each group had isolated pelvic injury and how many had accompanying limb fractures and urological injuries. Discussion: Please add study limitations: inter alia retrospective nature, relatively small group of patients. Please add the information that in the future a similar study should be performed on a larger group of patients.

Author Response

An interesting article, a topic that has not been fully understood so far.

However, I have a few comments:

INTRODUCTION: 

1.Please add a research hypothesis. 

Response: Thank you for your comment. Please see the revised manuscript on line 58-59

2.MATERIALS AND METHODS: Please state how many patients in each group had isolated pelvic injury and how many had accompanying limb fractures and urological injuries. 

Response: Thank you for your comments. In group 1, 4 patients had isolated pelvic injury and 17 patients had accompanying limb fracture or other associated injury such as urologic injury, pneumothorax, rectal injury, initial hypovolemic shock, abdominal injury, intracranial hemorrhage and chest injury. In group 2, 3 patients had isolated pelvic injury and 14 patients had accompanying limb fracture or other associated injury such as urologic injury, rectal injury, Morel-Lavallee lesion (internal degloving injury), chest injury, initial hypovolemic shock and head injury. Please see the revised manuscript on line 71-77

3.Discussion: Please add study limitations: inter alia retrospective nature, relatively small group of patients.

Response: Thank you for your comment. Please see the revised manuscript on line 283-286

4.Please add the information that in the future a similar study should be performed on a larger group of patients.

Response: Thank you for your comment. Please see the revised manuscript on line 283-286

Reviewer 3 Report

Topic: Sacral fractures in vertically unstable pelvic fractures are rare yet challenging to treat, with no clear consensus. It is a specialized topic among trauma surgeons. Therefore the topic is valuable to a narrow audience.

Method: if the same experienced pelvic trauma surgeon performed all surgeries, it is unfortunate that he decided to compare the techniques retrospectively.

How did you decide which technique to use for each patient? This is important due to a selection bias. The percutaneous technique was used more recent, on less severe cases, with less preoperative displacement or more severe injuries? You make no reference to selection of surgical procedure or comparison of patients before surgery. Did you have cases where the percutaneous reduction and fixation failed and was converted to traditional open technique?

How did you determine the cosmetic appearance?

Discussion: Rephrase the beginning – line 181 (‘most ideal’).

Fig 4 should be removed or moved to the Method/ surgical technique.  

You could comment on how to select cases for the percutaneous technique.

Overall: a good manuscript suitable for publication after revision.

Author Response

Topic: Sacral fractures in vertically unstable pelvic fractures are rare yet challenging to treat, with no clear consensus. It is a specialized topic among trauma surgeons. Therefore the topic is valuable to a narrow audience.

Method: if the same experienced pelvic trauma surgeon performed all surgeries, it is unfortunate that he decided to compare the techniques retrospectively.How did you decide which technique to use for each patient? This is important due to a selection bias. The percutaneous technique was used more recent, on less severe cases, with less preoperative displacement or more severe injuries? You make no reference to selection of surgical procedure or comparison of patients before surgery.

Response: Thank you for your comment. In our county, due to the different cost between theses implants, the pedicle screw and rod implantation by the minimally invasive method is more expensive and should be paid by the patient. But the pedicle screw and rod implantation by the open method could be supported by the National Health Insurance Administration after written application which should be reviewed and qualified by the expert. Actually, the implants and the surgical method were decided by the patients’ economic condition. This was the reason why these patients could not be randomized selected. The severity of fracture displacement or other accompanying injury are not the criteria for the patients’ group selection. Please see the revised manuscript on line 277-282

Did you have cases where the percutaneous reduction and fixation failed and was converted to traditional open technique?

Response: Thank you for your comment. In our study, no patient in the minimally invasive surgery group was converted to traditional open technique. Fortunately, all the patients in the group 2 had completed surgery without immediate complication or failed reduction or fixation. However, when we review our complication cases in the manuscript, we found there is a wrong description. In group 1, there were 2 complications, one was pedicle screw broken and the other was pedicle screw loosening. In group 2, there was one patient with wound infection over surgical wound of anterior pelvic ring fixation. We made revision in the revised manuscript on line 205-213

How did you determine the cosmetic appearance?

Response: Thanks for your comment. The cosmetic appearance is one of the questions in the Iowa pelvic score. It was determined by the patient’s feeling and feedback. We isolated the cosmetic appearance from the Iowa pelvic score because the total Iowa pelvic score showed no significant difference in both groups, however, if we analyzed the cosmetic appearance only, significant difference between two groups was noted.

Discussion: Rephrase the beginning – line 181 (‘most ideal’).

Response: We rephrased the beginning of discussion, thanks for your kindly suggestion. Please see the revised manuscript on line 215-216

Fig 4 should be removed or moved to the Method/ surgical technique.  

Response: Thanks for your comment. The Figure 4 was moved to the Method/ surgical technique as Figure 2. The previous Figure 2,3 was changed as Figure 3 and 4, respectively. Please see the revised manuscript on line 114-119, 167, 169 and 178

You could comment on how to select cases for the percutaneous technique.

Response: In our county, due to the different cost between theses implants, the pedicle screw and rod implantation by the minimally invasive method is more expensive and should be paid by the patient. But the pedicle screw and rod implantation by the open method could be supported by the National Health Insurance Administration after written application which should be reviewed and qualified by the expert. Actually, the implants and the surgical method were decided by the patients’ economic condition. This was the reason why these patients could not be randomized selected. The severity of fracture displacement or other accompanying injury are not the criteria for the patients’ group selection. Please see the revised manuscript on line 277-282

Overall: a good manuscript suitable for publication after revision.